# The Effect of Increasing the Proportion of Dietary Roughage Based on the Partial Replacement of Low-Quality Roughage with Alfalfa Hay on the Fatty Acid Profile of Donkey Milk

**DOI:** 10.3390/ani15030423

**Published:** 2025-02-03

**Authors:** Xiaoshuai Liang, Xiaoyu Guo, Yuanxi Yue, Fang Hui, Manman Tong, Yongmei Guo, Yaguang Zheng, Binlin Shi, Sumei Yan

**Affiliations:** Key Laboratory of Animal Nutrition and Feed Science, College of Animal Science, Inner Mongolia Agricultural University, Universities of Inner Mongolia Autonomous Region, Hohhot 010018, China; 18447052197@163.com (X.L.); gxy_2594@163.com (X.G.); yueyuanxi2013@163.com (Y.Y.); cf18447051697@163.com (F.H.); nndtmm@163.com (M.T.); ymguo2015@163.com (Y.G.); zhengyaguangg@163.com (Y.Z.); shibl@imau.edu.cn (B.S.)

**Keywords:** nutrition, diet, lactation, plasma, GC-MS

## Abstract

Donkey milk, as a dairy source, has a high polyunsaturated fatty acid profile, especially C18:3n3, which is significantly higher than cow’s milk, making it a low-fat product with a high content of functional fatty acids. The proportion and quality of dietary roughage affect the milk fatty acid profile of dairy animals. It was found that the substitution of alfalfa hay for part of the low-quality roughage and simultaneous reduction in the ratio of concentrate to roughage in the diet increased the C18:3n3 profile in the fatty acids of donkey’s milk and optimized the fatty acid profile of donkey milk.

## 1. Introduction

With the development of the diversified dietary structure of the population, consumers are paying more attention to the nutritional content of dairy milk. Many researchers have found a strong association between the composition of polyunsaturated fatty acids (PUFA) in dairy milk and health. n-3 polyunsaturated fatty acids (n-3 PUFA) are metabolic converters that activate lipolysis metabolism and inhibit lipogenesis [1] and have physiological functions such as enhancing intelligence [2], improving immunity [3], lowering blood lipids [4], and lowering blood pressure [5]. When saturated fatty acids (SFA)and n-6 polyunsaturated fatty acids (n-6 PUFA) are too high, it also affects the cholesterol (CHO) metabolism of the organism, leading to an increased risk of cardiovascular disease [6]. An imbalance in the n-6 polyunsaturated fatty acids (n-6 PUFA)/n-3 PUFA ratio (n-6/n-3) in the body may lead to physiological dysfunction or even produce disease. In human dietary unsaturated fatty acids, a lower n-6/n-3 ratio is better for health [7]. Since mammals cannot synthesize alpha-linolenic acid (ALA), the precursor of eicosapentaenoic acid(EPA) and docosahexaenoic acid(DHA), and C18:2n6, the precursor of C20:4n6 [8], they must intake them from exogenous sources. Donkey milk, as a dairy source, has a milk fat content of 1.3 ± 0.4% [9], and per 100 g of milk fat, the n-3 PUFA content of donkey milk is 7.97 g, and the ALA content is 7.25 g; the n-3 PUFA content of cow’s milk is 0.78 g, and the ALA content is 0.48 g [10], with ALA content of donkey milk fat being 15 times higher than that of cow’s milk. According to the recommendation of the WHO (1993) [11], the daily ALA requirement of an average person is 1000 mg. In every 100 mL of dairy products, donkey milk can provide 67.9 mg of ALA, while cow’s milk only has 17.5 mg [10]. This means that although the total fat content of donkey milk is lower than that of cow’s milk (3.6 ± 0.4%) [9], donkey milk is high in PUFA, especially in ALA, which is significantly higher than that of cow’s milk, and it is a low-fat product with a high content of functional fatty acids (FA).

FA in milk are mainly de novo synthesized by mammary epithelial cells and taken up from the blood; their composition and content are susceptible to dietary FA. The quality and source of roughage are significant factors influencing the FA composition of milk [12], and the use of high-quality roughage to replace some of the low-quality roughage to change the ratio of dietary concentrate to forage can affect the FA composition of milk, but at present, the relevant studies are mainly focused on ruminant dairy animals, and few data are available on equine animals, especially donkeys. Previous studies showed that fresh pasture fed to lactating donkeys instead of hay increased PUFA, n-3 PUFA content, and unsaturated fatty acids (UFA)/SFA (U/S) in donkey milk and decreased n-6/n-3 [13]. A change in feeding from 100% hay feeding to 50% hay and 50% pasture feeding increased the C18:3n3 and n-3 PUFA content of donkey milk [14]. The results of our previous studies found that appropriately increasing the dietary protein level [15] or using alfalfa hay to partially replace low-quality roughage to increase the proportion of roughage [16] would improve the digestion and utilization of nutrients as well as the metabolism of nitrogen in serum in lactating donkeys, which would enhance milk production performance. Due to the lack of depth of current research related to lactating donkeys, the effect of dietary nutrition on the milk FA profile of lactating donkeys is less reported, and it is not clear whether the ratio of dietary concentrates/roughage has an effect on the FA profile of donkey milk. Therefore, based on our previous studies, we hypothesized that increasing the proportion of dietary roughage based on the partial replacement of low-quality roughage with high-quality roughage could improve the FA profile of donkey milk at the same energy and protein levels of the ration, especially the composition of the PUFA profile. This study mainly investigated the effects of partial substitution of low-quality roughage with alfalfa hay and increasing the roughage proportion on the plasma and milk FA profile in lactating donkeys.

## 2. Materials and Methods

### 2.1. Experimental Design and Treatments

This study was performed at the Yulv Donkey Grassland Farm (Hohhot, China). All experimental procedures carried out on donkeys were conducted according to the national standard “Guidelines for Ethical Approval Document for Biomedical Research at Inner Mongolia Agricultural University. The experimental donkeys involved in this experimental project were examined by the Laboratory Animal Welfare and Ethics Committee of Inner Mongolia Agricultural University A sample of 16 Dezhou donkeys (n = 16) of the same age (6.73 ± 1.16 years), weight (250.84 ± 27.21 kg), parity (3.25 ± 1.12), and days in lactation (40 ± 4 days) were selected and divided on average into two treatment groups, the low alfalfa group (LG, roughage 60/concentrate 40) and the high alfalfa group (HG, roughage 70/concentrate 30) according to a one-way, totally randomized testing design. Alfalfa hay accounted for 44.85 g/kg of total dietary dry matter (DM) in LG, while that of HG was 179.48 g/kg of total dietary DM. Both diets had the same energy and protein levels. The energy value, expressed as digestible energy (DE), was calculated using the equations reported by INRA (2015) [17]. Neutral detergent fiber (NDF) was tested with heat-resistant amylase and expressed exclusive of residual ash. Acid detergent fiber (ADF) was expressed on residual ash only. The adaptation period to the diet was 1 week and the data collection period was 8 weeks.

Both groups of foals were kept with their dams, similar birth age (40 ± 4 days of age) and weight (52.38 ± 6.39 kg). Donkeys and foals were housed in two separate areas with peripheral paddocks (donkeys: 3.5 m × 2.0 m, foals: 1.6 m × 2.0 m) with sand on the floors inside and outside of the paddocks, as well as consistent temperature, humidity, and air circulation. Foals were separated from their dams from 7:00 to 10:00 a.m. and from 2:00 to 5:00 p.m. daily and spent the rest of the day with their dams in natural suckling. Diet composition, nutrient levels, and FA composition are shown in Table 1.

### 2.2. Feeding Management

A single donkey was housed in a single cubicle (3.5 m × 2.0 m) and given concentrate, corn silage, millet straw, and alfalfa at 7 a.m. and 2 p.m. daily. The millet straw was given five times a day. The previous day’s leftovers were removed before feeding in the morning. The amount of leftovers was recorded. Donkeys were separated from their foals for 3 h each time from 7:00 to 10:00 a.m. and 2:00 to 5:00 p.m., totaling 6 h per day for milking. Milking was performed daily at 10:00 a.m. and 5:00 p.m. using a separate vacuum milk extractor (Judu-H5402, Judu Technology Co. Ltd., Hebei, China). The vacuum pump had a vacuum level of around 50 KPa, a pulsation frequency of 60 ± 3 times per minute, and a pulsation ratio of 60:40. During milking, udders were cleaned with sterile wet wipes, and nipples with warm water and dried with sterile gauze. Milk yield was measured and recorded using a lactation meter connected to the milking machine.

Feed residues were collected each morning before the millet straw was fed, and the daily feeding rate was adjusted based on the previous day’s intake to achieve a target refusal rate of 10%. The specified proportions of concentrate and roughage were kept constant. The offered and rejected feeds were weighed daily, the DM content (AOAC, International, 2002, method 930.15) of the rations was determined, and the daily intake (DMI) was calculated for each group. Donkeys were weighted on the first and last mornings of the data collection period using a small weigh scale (Shunqiang, Suzhou, China) with fasting (× 12 h); initial and final body weights were recorded, and body weight loss was calculated for the data collection period. Body weight loss = (final body weight—initial body weight)/initial body weight.

Each livestock shed used in this trial was provided with four separate water bowls with floats, which were accessible to both foals and jennies, and the troughs, roughage racks, and concentrate troughs were cleaned on a daily basis.

### 2.3. Diets Samples and Analysis

Dietary samples were collected before the trial began, and those were divided into two parts. One sample was dried in an oven at 65 °C, crushed, sieved, and stored at −20 °C for the determination of conventional nutrients. This consists of the content of DM (AOAC, International, 2002, method 930.15), ether extract (EE) (AOAC, International, 2002, method 920.39), and hydrochloric acid insoluble ash (AIA) (AOAC, International, 2002, method 975.12) [18]. NDF and ADF were assayed following the method of Van Soest et al. (1991) [19]. Another sample was stored directly in an ultra-low temperature refrigerator at—80 °C for the determination of FA (AOAC, International, 2002, method 996.06) content [19]. FA profiles were calculated from the measurements, including: SCFA, short-chain fatty acid (4:0); MCFA, medium-chain fatty acid (6:0 + 8:0 + 10:0 + 11:0 + 12:0); LCFA, long-chain fatty acid (14:0 + 14:1 + 15:0 + 15:1 + 16:0 + 16:1 + 17:0 + 17:1 + 18:0 + 18:1nt9 + 18:1nc9 + 18:2nt6 + 18:2nc6 + 20:0 + 18:3n6 + 20:1 + 18:3n3 + 21:0 + 20:2n6 + 22:0 + 20:3n6 + 22:1 + 20:3n3 + 23:0 + 20:4n6 + 22:2n6 + 24:0 + 20:5n3 + 24:1 + 22:6n3). SFA (6:0 + 8:0 + 10:0 + 11:0 + 12:0 + 13:0 + 14:0 + 15:0 + 16:0 + 17:0 + 18:0 + 20:0 + 21:0 + 22:0 + 23:0 + 24:0); UFA, including MUFA: monounsaturated fatty acids and PUFA; MUFA, (14:1 + 15:1 + 16:1 + 17:1 + 18:1n9t + 18:1n9c + 20:1 + 22:1 + 24:1); PUFA: (n-6 PUFA + n-3 PUFA,); n-6 PUFA: (18:2nt6 + 18:2nc6 + 18:3n6 + 20:2n6 + 20:3n6 + 20:4n6 + 22:2n6); n-3PUFA: (18:3n3 + 20:3n3 + 20:5n3 + 22:6n3); n-6 LCPUFA, n-6 long chain polyunsaturated fatty acids (20:2n6 + 20:3n6 + 20:4n6 + 22:2n6); n-3 LCPUFA, n-3 long chain polyunsaturated fatty acids (20:3n3 + 20:5n3 + 22:6n3); n-6/n-3, n-6 PUFA/n-3 PUFA; U/S; UFA/SFA; P/S, PUFA/SFA. FA intake was determined according to the following equation: FA intake = DMI × dietary content of a single FA/metabolic weight (body weight ^0.75^).

### 2.4. Milk Sampling and Analysis

Milk was collected at weeks 1–8, preserved with preservative (D & F Control Systems Inc., Beverly, MA, USA), and maintained at 4 °C until analyzed. Milk fat content was determined using infrared spectrophotometry (Milkoscan FT +, Foss Analytical Co., Ltd., Hillerød, Denmark). Milk samples from each donkey were also collected on the last day of the eighth week, respectively, and the morning and afternoon samples were mixed in a 1:1 ratio poured evenly into a stainless steels tray, frozen at −80 °C, and transferred to a freezer dryer overnight. The following day, the samples were lyophilized in a Sublimator 3 × 4 × 5 lyophilizer (Zirbus Technology, Bad Grunt, Germany). The samples were placed in a condensation chamber under vacuum at a maximum pressure of 0.3 mbar, and the freeze dryer was operated at a maximum temperature of 35 °C. Lyophilization was completed in 72 h. The powders obtained were stored under vacuum in sealed polyethylene bags for FA analysis.

Before derivatization to FA methyl esters, milk samples were preprocessed by precipitation of protein and lipid extraction. The determination method referenced Nayak et al. (2020) [20]. The brief process is as follows: weigh lyophilized donkey milk powder into a hydrolysis tube, add 10 N potassium hydroxide and methanol solution, heat in a water bath, cool to room temperature, add 24 N sulfuric acid and mix, then heat again in a water bath and cool to room temperature, and then add hexane. After cooling to room temperature, add and vortex hexane, and transfer the remaining liquid to a centrifuge tube and centrifuge it, (1500× *g*). Transfer the supernatant to a centrifuge tube, add a small amount of anhydrous ammonium sulfate, vortex, and let stand. Aspirate the supernatant under a gentle stream of nitrogen and let it evaporate to dryness in a sample concentrator at 40 °C. Redissolve each residue in n-hexane and separate the shallow upper n-hexane phase using a syringe. Of the hexane extracted from the milk samples, 1 mL went directly to the gas chromatography followed by mass spectrometry (GC-MS).

Milk FA profile was determined as the FA methyl esters by GC-MS (7890A, Agilent Technologies Inc., Santa Clara, CA, USA). The chromatographic column was SPTM-2560 (100 m × 0.25 mm × 0.2 μm, Supelco Inc., Bellefonte, PA, USA). The programmed heating conditions were as follows: the initial temperature was 85 °C, maintained for 5 min; then it was increased to 120 °C at a rate of 5 °C/min, maintained for 5 min; increased to 180 °C at a rate of 5 °C/min, maintained for 5 min; then increased to 240 °C at a rate of 2 °C/min, maintained for 15 min; the inlet temperature was 248 °C; the nitrogen (N_2_) flow rate was 3 mL/min; and the shunt ratio was 1:9.

### 2.5. Blood Sampling and Analysis

On the last day of the trial, the blood sample from each donkey from the two treatment groups was collected preprandially (× 12 h fasting) in the morning. An ordinary blood collection tube and a sodium heparin blood collection tube (made by Corning, NY, USA) were used to collect blood by puncturing the jugular vein of the donkey through a disposable sterile blood collection needle (made by Yuehe, Shandong, China). The tubes were put on ice, allowed to stand, and then layered. They were promptly centrifuged at 4 °C and (2054.3× g) for 15 min. Serum and plasma were separated and stored at −80 °C until analysis. The FA profile in plasma was assessed using a meteorological chromatograph (GC-7890A Agilent Technologies, Santa Clara, CA, USA) [21]. The index of atherogenicity (IA) and index of thrombogenicity (IT) were calculated from the FA results using the equations described by Ulbricht and Southgate (1991) [22]:IA = (C12:0 + (4 × C14:0) + C16:0) (C12:0 + (4 × C14:0) + C16:0) (C12:0 + (4 × C14:0) + C16:0)/ΣMUFA+Σ (n − 6) + Σ (n − 3) ΣMUFA + Σ (n − 6) + Σ (n − 3) ΣMUFA + Σ (n − 6) + Σ(n − 3).IT = (C14:0 + C16:0 + C18:0) (C14:0 + C16:0 + C18:0) (C14:0 + C16:0 + C18:0)/0.5 × ΣMUFA + 0.5 × Σ(n − 6)+(3 × Σ(n − 3)) + (Σ(n − 3)/Σ(n − 6))0.5 × ΣMUFA + 0.5 × Σ (n − 6) + (3 × Σ (n − 3)) + (Σ (n − 3)/Σ (n − 6)) 0.5 × ΣMUFA + 0.5 × Σ (n − 6) + (3 × Σ (n − 3)) + (Σ (n − 3)/Σ (n − 6))

The same parameters were determined for donkey milk. Serum concentrations of CHO, β-hydroxybutyric acid (D_3_HB), triglycerides (TG), high-density lipoprotein cholesterol (HDL-C), low-density lipoprotein cholesterol (LDL-C), and nonesterified fatty acids (NEFA) were measured using a double-antibody sandwich enzyme-linked immunosorbent assay (ELISA) kit supplied by Boxin Bio (Lepu Medical Equipment Co., Ltd., Beijing, China).

### 2.6. Statistical Analysis

All statistical analyses were performed by SAS (2016). The test unit was the individual donkey. Body weight loss, FA intake, serum biochemical indices, and FA profiles in plasma and milk were analyzed by paired samples *t*-test. The data are expressed as the mean of the least squares and the standard error of the mean. Differences were considered significant at *p* ≤ 0.10.

## 3. Results

### 3.1. Production Performance and Fatty Acid Intake

The effects of alfalfa replacement of low roughage on lactation performance and FA intake of donkeys are shown in Table 2. Compared with the LG, milk yield was higher in the HG (*p* = 0.042), and the body weight loss, DMI, and milk fat content were not different between the two groups. The fed intake of C13:0, C15:0, C23:0, C17:1, C24:1, n-6 PUFA (except C18:2nc6), n-3 PUFA (except C22:6n3), SFA, SCFA, MCFA, LCFA, n-6 LCPUFA, n-3 LCPUFA, U/S, and P/S in the HG was higher than that in the LG (*p* < 0.10). The fed intake of C4:0, C8:0, C12:0, C16:1, C20:1, C18:2nc6, MUFA, n-6/n-3 ratio, and IA was lower in the HG than in the LG (*p* < 0.05). Except for the above FA, the remaining FA intake did not differ between the two groups (*p* > 0.10).

### 3.2. Fatty Acid Profiles of Plasma and Milk

According to the results in Table 3, in plasma FA profile, compared to the LG, the plasma profiles of C22:0, C24:0, C18:3n3, C22:1, C18:2nt6, C20:2n6, n-3 PUFA, n-6 LCPUFA, and LCFA were increased in the HG (*p* < 0.10); the profiles of C4:0, C6:0, C11:0, and C14:1 were decreased in the HG (*p* < 0.05); the profile of MCFA, IA, and the n-6/n-3 ratio was decreased in the HG (*p* = 0.051, *p* = 0.092 and *p* = 0.073).

According to the results in Table 4, in milk FA profiles, compared to the LG, the profiles of C18:1t9, C20:0, C22:1, C18:3n3, LCFA, PUFA, and n-3 PUFA were increased in the HG (*p* < 0.10); the profile of C20:1, n-6/n-3 ratio, and IT were decreased in the HG (*p* = 0.024; *p* = 0.019 and *p* = 0.002). Other FA profiles did not differ between the two groups (*p* > 0.10).

### 3.3. Biochemical Parameters in the Serum

As shown in Table 5, serum concentrations of CHO and LDL-C were reduced in the HG versus the LG (*p* = 0.03 and *p* < 0.01, respectively), as were concentrations of D_3_HB (*p* = 0.08). There were no significant differences between the two groups in the concentrations of TG, HDL-C, or NEFA in the serum (*p* > 0.10).

## 4. Discussion

At present, detailed information on the mechanism of milk FA synthesis in lactating donkeys and its influencing factors has not been reported. Most studies have shown that changing the quality or source of dietary roughage or dietary concentrate to coarse ratio can affect the composition of plasma and milk FA in dairy animals. Previously, it was found that semi-grazing (pasture 50% and hay 50%) feeding increased C18:1c9, C18:3n-3, and n-3 PUFA content in donkey milk compared to housed feeding (pasture 0%, hay 90%, and cereal mix 10%) [14]. The replacement of hay with fresh forage in the ration increased the PUFA, n-3 PUFA content, and U/S ratio and decreased the n-6/n-3 ratio in donkey milk [13]. When the roughage composition of the diet is increased, the fermentation products have a higher proportion of acetic acid [23], which is a precursor for fat synthesis in the mammary gland [24], facilitating the synthesis of milk fat, increasing donkey milk fat production or optimizing milk FA composition. In the present experiment, under the conditions of equal energy and nitrogen in the ration, increasing the percentage of dietary roughage from 60% to 70%, the FA profile in the milk of the HG had a similar pattern to the FA profile of plasma. When alfalfa hay partially replaced the low-quality roughage fed to jennies, there was a tendency for the ratios of LCFA, n-3 PUFA, and n-3 LCPUFA in donkey milk to increase, with a significant increase in the ratios of ALA, C181nt9, and C22:1 in particular. Probably due to the increased percentage of roughage due to the partial replacement of millet straw by alfalfa hay in the diet, the increased intake of LCFA in donkeys, the increased plasma LCFA concentration during lactation, and the mobilization of plasma LCFA by the mammary gland for transfer to the milk resulted in an increase in the concentration of LCFA in the milk.

Studies have confirmed that most of the remaining LCFA in milk, excluding SCFA and MCFA synthesized de novo in the mammary gland, comes from LCFA taken up by the mammary gland from the blood [25], which is mainly derived from the diet [26]. This shows that the diet FA composition not only affects plasma FA profile, but plasma FA plays an important role in the transition from diet FA to milk FA and is an important hub linking diet FA and milk FA. PUFA are enriched in oilseeds and are also seen in green forage grasses, and humans do not possess the required desaturase enzymes to carry out sequential elongation and desaturation reactions of multiple FA by oleic acid to produce LCPUFA with specific functions [27]. Millet straw is cheap and plentiful and is the main feed for donkey forage diets, but it has a very low nutritive value. Increasing the proportion of millet straw in the diet can reduce the cost of diet, but it requires an increase in the proportion of concentrate feeds or an improvement in roughage quality to maintain the same nutrient level. Studies have shown that alfalfa has a higher PUFA content than millet straw [28], so increasing the proportion of alfalfa in roughage or increasing the proportion of total roughage can increase the total PUFA content of the diet. Some studies have demonstrated that in vitro degradation of nutrients and fermentation of cecal fluid can be improved by replacing 20% of a mixture of millet straw and maize stover with alfalfa hay in the substrate using in vitro methods [29]; alfalfa hay had the highest in vitro degradation of crude proteins in the donkey stomach–small intestines–caecum, relative to millet straw, corn straw, oatgrass, and goatgrass [30]. Donkeys are characterized by monogastric hindgut digestion [31], and roughage provides energy through volatile fatty acids generated by cecal and colonic microbial fermentation [32], which are used for activities such as life support and lactation. Previously in our study, it was also found that alfalfa hay partially replacing low-quality roughage increased the digestion and utilization of dietary energy in lactation donkeys [16]. This indirectly explains that the synthesis of PUFA provided sufficient energy in the milk of donkeys in the HG group, and it may also be related to the fact that alfalfa grass promotes nutrient digestion and absorption and alters the intestinal microbial ecology and digestive enzyme activity in the hindgut digestive tract, which is less studies lactating donkeys at this stage and requires in-depth studies and analyses. It has also been reported that decreasing the concentration of C16:0 and increasing the concentration of C18:1 in the ration of early lactation cows improves the body condition score of cows after giving birth, and there was no difference in body weight gain at 21 days postpartum [33]. This experiment found that alfalfa hay replacing part of low-quality roughage and increasing the proportion of roughage had no significant effect on the body weight loss of lactating donkeys, although LG displays higher values. There are fewer reports on lactating donkeys, and the reasons need to be further analyzed.

The present study found that increasing the proportion of roughage by increasing the alfalfa hay content in the diet from 44.85 to 179.48 g/kg (DM) in place of some millet straw could increase the intake of LCFA, n-3 PUFA, ALA, and n-6 LCPUFA in the diet of lactating donkeys. In plasma, the profiles of MCFA and LCFA increased; and the profiles of n-3 PUFA and n-6 LCPUFA also tended to increase, with a significant increase in the profiles of ALA and C20:2n6. C20:2n6 is synthesized by linoleic acid as a precursor through carbon chain extension [27]. It could partially explain the elevated C20:2n6 in the blood of HG. n-3 PUFA and ALA were increased in the donkeys of the HG although there was no difference in DMI between the two groups, which was related to the higher content of n-3 PUFA and n-6 LCPUFA in the HG diets. It has been suggested that mammary epithelial cells are able to utilize exogenously ingested LCFA to synthesize new triacylglycerols composed of different types of FA, which are accumulated and secreted into the milk as lipid droplets [34], thus rendering the composition of FA in the milk susceptible to the FA composition of the diet. This could partly explain the elevated n-3 PUFA in the milk of the HG group, which may be related to the ability of mammary cells to take up FA from the blood. Currently, fewer studies have been reported in this area in lactating donkeys, and later analysis of blood metabolites and the activity of related enzymes need to be analyzed in order to further explore the mechanisms.

The two classes of essential PUFA are the n-6 and n-3 types. Linoleic acid in n-6 PUFA is a precursor of arachidonic acid from which biologically active enediones such as the prostaglandin subfamily are derived. The ALA in n-3 PUFA is converted to EPA and DHA by elongases and desaturases [35]. The n-3 PUFA are essential nutrients for the human body and have positive effects on human health and neurodevelopment of children [36], as well as the prevention and treatment of cardiovascular diseases, tumors, diabetes, and kidney diseases [37]. Among them, EPA and DHA have the effect of reducing the levels of TG, CHO, and LDL-C in human blood, thus improving blood circulation and reducing blood viscosity [38]. The content of SFA in livestock products, especially C14:0 and C16:0, can increase the level of LDL-C in human blood [39], and changes in LDL-C and HDL-C concentrations serve as a judgmental indicator of the causes of cardiovascular disease in humans. AI and TI are commonly used as reliable biomarkers to predict atherosclerosis and cardiovascular disease events in current research. In this study, we found that the increase in the alfalfa hay content in the diet from 44.85 to 179.48 g/kg (DM) in place of some millet straw and increasing the percentage of roughage in the diet from 60% to 70% significantly reduced the concentration of serum CHO, LDL-C, and D_3_HB and reduced AI in plasma in donkeys. A higher proportion of n-6/n-3 in the diet also acts as a potential risk factor for cardiovascular disease [40]. The present study showed that increasing alfalfa hay in lactating donkeys increased the profiles of all functional FA in donkey milk and decreased the ratio of n-6/n-3; the content of C14:0 and C16:0 in the LG was not different from that of the HG. According to Simopoulos (2002), an n-6/n-3 ratio below 2.0 is highly desirable and reduces the risk of many chronic diseases [41]. In our study, the mean n-6/n-3 ratio in both groups was 5.73, which was higher than the desirable ratio due to the high proportion of n-6 PUFA in donkey milk, but the partial replacement of low-quality roughage by alfalfa hay still reduced the n-6/n-3 ratio in donkey milk. An et al. (2023) showed that milk and dairy products exhibited IA in the range of 1.67–4.32 and IT in the range of 2.34–5.76 [42]. The mean values of IA and IT for donkey milk in both groups in the trial (IA: 1.40; IT: 1.41) were within this range and were similar to the results reported for mare’s milk by Czyżak-Runowska et al. (2021) (IA: 0.93–1.38; IT: 0.59–0.69) [43]. This indicates that the quality and proportion of roughage have an important effect on the FA composition of donkey milk and that the FA composition of donkey milk will be optimized after reducing the level of concentrate and replacing low-quality roughage with alfalfa. In addition to the concentrate-to-roughage ratio, there were some differences in the composition of other ingredients between the two diets, which needed to be further explored in order to adjust for equal energy and protein and to understand whether this also affects the composition of n-3 PUFA and n-6 PUFA in donkey milk.

In addition, the n-3 PUFA content of donkey milk was 7.97 g and ALA content was 7.25 g per 100 g of milk fat; the n-3 PUFA content of cow milk was 0.78 g and ALA content was 0.48 g [10], and the ALA content of donkey milk fat was 15 times higher than that of cow milk. According to previous reports, the content of ALA in cow’s milk fat was found to be about 0.29–0.79 g [44,45,46], and in combination with the present study, the content of ALA in donkey milk fat was found to be about 3.13–8.76 g [10,47,48]. Donkey milk has a higher content of UFA, especially ALA, than cow’s milk and, therefore, remains the main dairy source of n-3 PUFA for the human body.

## 5. Conclusions

The partial replacement of low-quality roughage with alfalfa hay and increasing the proportion of roughage in the diet could increase the ALA fatty acid content, with a tendency to increase the n-3 PUFA and LCFA content and decrease the n-6/n-3 and IT of donkey milk. Those optimized the milk fatty acid quality.

## Figures and Tables

**Table 1 animals-15-00423-t001:** The experimental diet composition, nutrient level, and fatty acid content.

Items	Treatment	
	LG	HG
Feed ingredients, g/kg DM	concentrate/roughage ratio	
	40:60	30:70
Roughage		
Alfalfa hay ^1^	44.85	179.48
Millet straw ^2^	437.10	384.60
Corn silage ^3^	111.93	130.87
Concentrate		
Corn	183.46	159.67
Soybean meal	91.77	71.91
Corn gluten meal	17.94	18.60
Distillers dried grains with solubles	9.26	4.63
Wheat bran	37.38	14.96
Corn germ meal	43.21	15.48
NaHCO_3_	3.27	0.00
NaCl	4.03	4.03
CaCO_3_	3.59	3.59
CaHPO_4_	7.18	7.19
Premix ^4^	5.00	5.00
Nutrient composition, g/kg DM		
Energy content ^5^, MJ/kg	12.46	12.43
Dry matter, DM, g/kg	872.10	872.20
Crude protein	140.32	140.81
Ether extract	24.70	24.30
aNDFom	530.50	548.90
ADFom	324.50	345.30
Fatty acid content ^6^, g/kg DM		
Saturated fatty acids		
C4:0	0.007	0.007
C 6:0	0.010	0.010
C8:0	0.010	0.007
C10:0	0.025	0.022
C11:0	0.022	0.022
C12:0	0.195	0.179
C13:0	0.010	0.010
C14:0	0.190	0.192
C15:0	0.077	0.097
C16:0	5.636	5.822
C17:0	0.158	0.165
C18:0	1.067	1.018
C20:0	0.348	0.328
C21:0	0.047	0.040
C22:0	0. 232	0.245
C23:0	0.161	0.214
C24:0	0.291	0.306
Monounsaturated fatty acids		
C14:1	0.015	0.015
C15:1	0.007	0.007
C16:1	0.023	0.022
C17:1	0.014	0.014
C18:1nt9	0.047	0.049
C18:1nc9	4.016	4.041
C20:1	0.062	0.051
C22:1	0.042	0.046
C24:1	0.015	0.017
n-6 Polyunsaturated fatty acids		
C18:2nt6	0.007	0.010
C18:2nc6	8.852	8.160
C18:3n6	0.032	0.034
C20:2n6	0.025	0.041
C20:3n6	0.005	0.007
C20:4n6	0.124	0.148
C22:2n6	0.015	0.019
n-3 Polyunsaturated fatty acids		
C18:3n3	2.979	3.222
C20:3n3	0.025	0.044
C20:5n3	0.094	0.117
C22:6n3	0.032	0.034
Sum and Ratio		
SCFA	0.007	0.005
MCFA	0.245	0.202
LCFA	24.448	24.093
SFA	8.593	7.776
UFA	16.107	16.524
MUFA	4.076	4.532
PUFA	12.031	11.994
n-3 PUFA	3.132	3.417
n-6 PUFA	8.899	8.578
n-3 LCPUFA	0.151	0.192
n-6 LCPUFA	0.17	0.216
n-6/n-3	0.003	0.003
U/S	1.874	2.125
P/S	1.400	1.542

Abbreviations: LR, low alfalfa hay group, 40 concentrate/60 roughage, alfalfa hay: 44.85g/kg dry matter; HR, high alfalfa hay group, 30 concentrate/70 roughage alfalfa hay: 179.48 g/kg dry matter. ^1^ Nutrient content (g/kg of DM): CP, 181.9; aNDFom, 538.9; ADFom, 397.8. ^2^ Nutrient content (g/kg of DM): CP, 54.7; aNDFom, 674.2; ADFom, 425.2. ^3^ Nutrient content (g/kg of DM): CP, 86.0; aNDFom, 501.7; ADFom, 264.8. ^4^ The treatment diet of premix contained (g/kg of DM) a maximum of vitamin A 6000 IU, vitamin D 1250 IU, vitamin E 15 IU, Fe 40 mg, Cu 8 mg, Zn 60 mg, Mn 60 mg, I 0.36 mg, Se 0.30 mg, and Co 0.50 mg. ^5^ Digestible energy was a calculated value. ^6^ Fatty acid composition: SCFA, short-chain fatty acid (4:0); MCFA, medium-chain fatty acid (6:0 + 8:0 + 10:0 + 11:0 + 12:0); LCFA, long-chain fatty acid (14:0 + 14:1 + 15:0 + 15:1 + 16:0 + 16:1 + 17:0 + 17:1 + 18:0 + 18:1nt9 + 18:1nc9 + 18:2nt6 + 18:2nc6 + 20:0 + 18:3n6 + 20:1 + 18:3n3 + 21:0 + 20:2n6 + 22:0 + 20:3n6 + 22:1 + 20:3n3 + 23:0 + 20:4n6 + 22:2n6 + 24:0 + 20:5n3 + 24:1 + 22:6n3). SFA, saturated fatty acids (6:0 + 8:0 + 10:0 + 11:0 + 12:0 + 13:0 + 14:0 + 15:0 + 16:0 + 17:0 + 18:0 + 20:0 + 21:0 + 22:0 + 23:0 + 24:0); UFA, unsaturated fatty acids include MUFA: monounsaturated fatty acids and PUFA: polyunsaturated fatty acids; MUFA, (14:1 + 15:1 + 16:1 + 17:1 + 18:1n9t + 18:1n9c + 20:1 + 22:1 + 24:1); PUFA: (n-6 PUFA, polyunsaturated fatty acids + n-3 PUFA, n-3 polyunsaturated fatty acids); n-6 PUFA: (18:2nt6 + 18:2nc6 + 18:3n6 + 20:2n6 + 20:3n6 + 20:4n6 + 22:2n6); n-3PUFA: (18:3n3 + 20:3n3 + 20:5n3 + 22:6n3); n-6 LCPUFA, n-6 long chain polyunsaturated fatty acids (20:2n6 + 20:3n6 + 20:4n6 + 22:2n6); n-3 LCPUFA, n-3 long chain polyunsaturated fatty acids (20:3n3 + 20:5n3 + 22:6n3); n-6/n-3, n-6 PUFA/n-3 PUFA; U/S; UFA/SFA; P/S, PUFA/SFA.

**Table 2 animals-15-00423-t002:** Production performance and fatty acid intake of lactating donkeys fed different quality roughage diets.

	Treatment			
Items	LG	HG	SEM	*p*-Value
Production performance ^1^				
Body weight loss, %	6.43	6.19	0.649	0.789
Dry matter intake, kg/day	7.02	7.05	0.031	0.757
Milking yield, kg/day	0.78	0.97	0.001	<0.001
Milk fat, g/kg	3.45	3.21	0.016	0.175
Saturated fatty acids ^2^, mg/kg W^0.75 ×^ day				
C4:0	0.61	0.55	0.015	0.021
C6:0	0.84	0.79	0.030	0.153
C8:0	0.78	0.66	0.017	<0.001
C10:0	2.15	2.19	0.061	0.869
C11:0	1.98	1.62	0.122	0.144
C12:0	16.62	15.32	0.369	0.039
C13:0	0.74	0.94	0.021	0.006
C14:0	16.00	16.30	0.381	0.609
C15:0	6.49	8.22	0.199	<0.001
C16:0	476.94	487.32	12.457	0.601
C17:0	13.33	13.89	0.353	0.326
C18:0	90.32	85.22	2.231	0.164
C20:0	29.39	27.43	0.720	0.103
C21:0	4.05	4.06	0.105	0.972
C22:0	19.65	20.46	0.520	0.336
C23:0	13.54	18.12	0.425	<0.001
C24:0	24.69	25.54	0.646	0.417
Monounsaturated fatty acids, mg/kg W^0.75 ×^ day				
C14:1	1.22	1.29	0.032	0.219
C15:1	0.60	0.62	0.023	0.410
C16:1	19.87	18.17	0.478	0.042
C17:1	1.20	1.30	0.034	0.086
C18:1nt9	4.03	4.00	0.102	0.858
C18:1nc9	339.76	343.80	8.873	0.774
C20:1	5.21	4.36	0.123	<0.001
C22:1	3.60	3.80	0.104	0.202
C24:1	1.22	1.42	0.038	0.006
n-6 Polyunsaturated fatty acids, mg/kg W^0.75 ×^ day				
C18:2nt6	0.71	0.83	0.020	0.012
C18:2nc6	749.02	683.03	18.073	0.037
C18:3n6	2.65	3.00	0.780	0.019
C20:2n6	2.18	3.48	0.080	<0.001
C20:3n6	0.47	0.53	0.013	0.013
C20:4n6	10.38	12.37	0.303	0.003
C22:2n6	1.29	1.63	0.227	0.001
n-3 Polyunsaturated fatty acids, mg/kg W^0.75 ×^ day				
C18:3n3	174.98	210.13	5.063	0.010
C20:3n3	2.02	3.65	0.082	<0.001
C20:5n3	8.02	9.71	0.237	0.002
C22:6n3	2.81	2.80	0.072	0.907
Sum and Ratio, mg/kg W^0.75 ×^ day				
SCFA	0.46	0.49	0.001	0.002
MCFA	16.40	20.11	0.489	0.001
LCFA	2026.70	2190.70	45.883	0.037
SFA	664.40	719.41	18.441	0.089
UFA	1421.40	1348.20	35.455	0.211
MUFA	389.75	335.51	9.039	0.003
PUFA	1031.60	1007.10	26.257	0.557
n-3 PUFA	264.95	290.66	7.331	0.034
n-6 PUFA	766.69	704.66	18.597	0.053
n-3 LCPUFA	12.85	16.16	0.391	<0.001
n-6 LCPUFA	14.33	18.02	0.436	<0.001
n-6/n-3	16.44	12.79	0.356	<0.001
U/S	5.64	4.84	0.131	0.002
P/S	4.00	3.57	0.096	0.016
IA	0.87	0.79	0.021	0.024
IT	0.18	0.15	0.011	0.104

Abbreviations: LR, low alfalfa group, 40 concentrate/60 roughage, alfalfa hay: 44.85g/kg dry matter; HR, high alfalfa group, 30 concentrate/70 roughage alfalfa hay: 179.48g/kg dry matter; DMI, dry matter intake; SEM, standard error of the mean. ^1^ In production performance, body weight loss = (final body weight—initial body weight)/initial body weight; DMI, milking yield, and milk fat data are referenced from Liang et al. [16]. ^2^ Fatty acid composition: SCFA, short-chain fatty acid (4:0); MCFA, medium-chain fatty acid (6:0 + 8:0 + 10:0 + 11:0 + 12:0); LCFA, long-chain fatty acid (14:0 + 14:1 + 15:0 + 15:1 + 16:0 + 16:1 + 17:0 + 17:1 + 18:0 + 18:1nt9 + 18:1nc9 + 18:2nt6 + 18:2nc6 + 20:0 + 18:3n6 + 20:1 + 18:3n3 + 21:0 + 20:2n6 + 22:0 + 20:3n6 + 22:1 + 20:3n3 + 23:0 + 20:4n6 + 22:2n6 + 24:0 + 20:5n3 + 24:1 + 22:6n3). SFA, saturated fatty acids (6:0 + 8:0 + 10:0 + 11:0 + 12:0 + 13:0 + 14:0 + 15:0 + 16:0 + 17:0 + 18:0 + 20:0 + 21:0 + 22:0 + 23:0 + 24:0); UFA, unsaturated fatty acids including MUFA: monounsaturated fatty acids and PUFA: polyunsaturated fatty acids; MUFA: (14:1 + 15:1 + 16:1 + 17:1 + 18:1n9t + 18:1n9c + 20:1 + 22:1 + 24:1); PUFA: (n-6 PUFA, n-6 polyunsaturated fatty acids + n-3 PUFA, n-3 polyunsaturated fatty acids); n-6 PUFA: (18:2nt6 + 18:2nc6 + 18:3n6 + 20:2n6 + 20:3n6 + 20:4n6 + 22:2n6); n-3PUFA: (18:3n3 + 20:3n3 + 20:5n3 + 22:6n3); n-6 LCPUFA, n-6 long chain polyunsaturated fatty acids (20:2n6 + 20:3n6 + 20:4n6 + 22:2n6); n-3 LCPUFA, n-3 long chain polyunsaturated fatty acids (20:3n3 + 20:5n3 + 22:6n3); n-6/n-3, n-6 PUFA/n-3 PUFA; U/S; UFA/SFA; P/S, PUFA/SFA; IA, index of atherogenicity; IT, index of thrombogenicity; fatty acid intake = DMI × dietary content of a single fatty acid/metabolic weight (body weight ^0.75^).

**Table 3 animals-15-00423-t003:** Fatty acid profiles of plasma of lactating donkeys fed different quality roughage diets.

	Treatment			
FAME (g/100 g) ^1^	LG	HG	SEM	*p*-Value
Saturated fatty acids				
C4:0	0.16	0.10	0.011	0.005
C 6:0	0.08	0.06	0.005	0.028
C8:0	0.03	0.04	0.003	0.338
C10:0	0.08	0.08	0.008	0.953
C11:0	0.11	0.07	0.009	0.018
C12:0	0.13	0.11	0.01	0.165
C13:0	0.12	0.11	0.012	0.914
C14:0	0.58	0.44	0.045	0.104
C15:0	0.16	0.15	0.01	0.399
C16:0	16.72	15.95	0.342	0.115
C17:0	0.57	0.59	0.048	0.819
C18:0	18.23	18.66	0.419	0.468
C20:0	0.59	0.59	0.019	0.893
C21:0	0.05	0.06	0.004	0.347
C22:0	0.03	0.26	0.014	<0.001
C23:0	0.06	0.05	0.007	0.235
C24:0	0.04	0.06	0.004	0.014
Monounsaturated fatty acids				
C14:1	0.12	0.08	0.011	0.025
C15:1	0.16	0.16	0.010	0.804
C16:1	0.98	0.89	0.068	0.406
C17:1	0.03	0.03	0.003	0.841
C18:1t9	0.14	0.12	0.016	0.391
C18:1c9	13.49	12.77	0.552	0.446
C20:1	0.32	0.35	0.024	0.509
C22:1	4.56	5.49	0.307	0.052
C24:1	0.40	0.40	0.045	0.939
n-6 Polyunsaturated fatty acids				
C18:2nt6	0.23	0.29	0.021	0.058
C18:2nc6	37.36	38.32	0.953	0.564
C18:3n6	0.92	0.89	0.074	0.749
C20:2n6	0.45	0.58	0.033	0.019
C20:3n6	0.22	0.22	0.014	0.947
C20:4n6	0.63	0.70	0.027	0.168
C22:2n6	0.17	0.17	0.013	0.966
n-3 Polyunsaturated fatty acids				
C18:3n3	0.73	0.89	0.042	0.026
C20:3n3	0.13	0.17	0.016	0.108
C20:5n3	0.08	0.08	0.012	0.983
C22:6n3	0.20	0.22	0.043	0.787
Sum and Ratio				
SCFA	0.13	0.11	0.012	0.508
MCFA	0.36	0.29	0.02	0.051
LCFA	99.45	99.61	0.065	0.002
SFA	37.34	37.36	0.413	0.966
UFA	62.66	62.68	0.414	0.976
MUFA	21.42	21.11	0.832	0.824
PUFA	41.21	42.37	0.682	0.296
n-3 PUFA	1.13	1.34	0.019	0.012
n-6 PUFA	40.5	40.31	0.628	0.825
n-3 LCPUFA	0.41	0.48	0.044	0.364
n-6 LCPUFA	1.50	1.63	0.051	0.097
n-6/n-3	33.85	30.08	1.367	0.073
U/S	1.64	1.68	0.017	0.22
P/S	1.09	1.11	0.015	0.372
IA	0.29	0.27	0.007	0.092
IT	0.46	0.45	0.008	0.793

Fatty acid methyl esters, g per 100 g of fatty acid. Abbreviations: treatment: LR, low alfalfa hay group, 40 concentrate/60 roughage, alfalfa hay: 44.85g/kg dry matter; HR, high alfalfa hay group, 30 concentrate/70 roughage alfalfa hay: 179.48g/kg dry matter; SEM, standard error of the mean. ^1^ Fatty acid composition: SCFA, short-chain fatty acid (4:0); MCFA, medium-chain fatty acid (6:0 + 8:0 + 10:0 + 11:0 + 12:0); LCFA, long-chain fatty acid (14:0 + 14:1 + 15:0 + 15:1 + 16:0 + 16:1 + 17:0 + 17:1 + 18:0 + 18:1nt9 + 18:1nc9 + 18:2nt6 + 18:2nc6 + 20:0 + 18:3n6 + 20:1 + 18:3n3 + 21:0 + 20:2n6 + 22:0 + 20:3n6 + 22:1 + 20:3n3 + 23:0 + 20:4n6 + 22:2n6 + 24:0 + 20:5n3 + 24:1 + 22:6n3). SFA, saturated fatty acids (6:0 + 8:0 + 10:0 + 11:0 + 12:0 + 13:0 + 14:0 + 15:0 + 16:0 + 17:0 + 18:0 + 20:0 + 21:0 + 22:0 + 23:0 + 24:0); UFA, unsaturated fatty acids including MUFA: monounsaturated fatty acids and PUFA: polyunsaturated fatty acids; MUFA: (14:1 + 15:1 + 16:1 + 17:1 + 18:1n9t + 18:1n9c + 20:1 + 22:1 + 24:1); PUFA: (n-6 PUFA, n-6 polyunsaturated fatty acids + n-3 PUFA, n-3 polyunsaturated fatty acids); n-6 PUFA: (18:2nt6 + 18:2nc6 + 18:3n6 + 20:2n6 + 20:3n6 + 20:4n6 + 22:2n6); n-3PUFA: (18:3n3 + 20:3n3 + 20:5n3 + 22:6n3);; n-6 LCPUFA, n-6 long chain polyunsaturated fatty acids (20:2n6 + 20:3n6 + 20:4n6 + 22:2n6); n-3 LCPUFA, n-3 long chain polyunsaturated fatty acids (20:3n3 + 20:5n3 + 22:6n3); n-6/n-3, n-6 PUFA/n-3 PUFA; U/S; UFA/SFA; P/S, PUFA/SFA; IA, index of atherogenicity; IT, index of thrombogenicity.

**Table 4 animals-15-00423-t004:** Fatty acid profiles of milk of lactating donkeys fed different quality roughage diets.

	Treatment			
FAME (g/100 g) ^1^	LG	HG	SEM	*p*-Value
Saturated fatty acids				
C4:0	0.02	0.01	0.003	0.744
C 6:0	0.05	0.05	0.012	0.913
C8:0	4.13	3.86	0.241	0.446
C10:0	11.69	11.24	0.552	0.574
C11:0	0.03	0.03	0.003	1.000
C12:0	11.19	10.55	0.864	0.611
C13:0	0.04	0.03	0.004	0.555
C14:0	8.57	8.35	0.872	0.869
C15:0	0.43	0.38	0.035	0.319
C16:0	23.64	22.18	1.440	0.483
C17:0	1.18	1.28	0.059	0.336
C18:0	1.92	2.08	0.093	0.308
C20:0	0.06	0.07	0.003	0.087
C21:0	0.12	0.12	0.011	1.000
C22:0	0.03	0.04	0.004	0.192
C23:0	0.01	0.01	0.002	0.363
C24:0	0.01	0.02	0.003	0.798
Monounsaturated fatty acids				
C14:1	0.36	0.33	0.036	0.499
C15:1	0.01	0.01	0.001	0.391
C16:1	2.86	2.69	0.248	0.636
C17:1	0.39	0.38	0.043	0.846
C18:1t9	0.08	0.17	0.013	<0.001
C18:1c9	20.10	18.70	0.777	0.286
C20:1	0.32	0.20	0.032	0.024
C22:1	0.07	0.10	0.010	0.042
C24:1	0.02	0.02	0.003	0.398
n-6 Polyunsaturated fatty acids				
C18:2nt6	0.03	0.04	0.003	0.231
C18:2nc6	15.87	18.14	1.010	0.153
C18:3n6	0.02	0.02	0.002	0.800
C20:2n6	0.48	0.52	0.061	0.678
C20:3n6	0.06	0.06	0.004	0.959
C20:4n6	0.09	0.10	0.009	0.614
C22:2n6	0.03	0.03	0.003	0.302
n-3 Polyunsaturated fatty acids				
C18:3n3	2.54	3.71	0.361	0.048
C20:3n3	0.08	0.10	0.015	0.195
C20:5n3	0.01	0.02	0.003	0.694
C22:6n3	0.02	0.02	0.002	0.959
Sum and Ratio				
SCFA	0.02	0.01	0.001	0.684
MCFA	24.13	23.06	1.053	0.979
LCFA	71. 54	77.13	1.482	0.057
SFA	58.16	55.47	1.868	0.333
UFA	41.84	44.53	1.687	0.333
MUFA	23.56	21.95	0.593	0.135
PUFA	18.82	24.21	1.640	0.085
n-3 PUFA	2.65	3.85	0.371	0.058
n-6 PUFA	16.46	18.80	1.044	0.417
n-3 LCPUFA	0.11	0.14	0.009	0.088
n-6 LCPUFA	0.73	0.74	0.072	0.909
n-6/n-3	6.27	5.19	0.352	0.019
U/S	0.80	0.81	0.083	0.950
P/S	0.35	0.42	0.036	0.275
IA	1.47	1.32	0.064	0.173
IT	1.55	1.26	0.035	0.002

Abbreviations: LR, low alfalfa hay group, 40 concentrate/60 roughage, alfalfa hay: 44.85g/kg dry matter; HR, high alfalfa hay group, 30 concentrate/70 roughage alfalfa hay: 179.48g/kg dry matter; SEM, standard error of the mean. ^1^ Fatty acid composition: SCFA, short-chain fatty acid (4:0); MCFA, medium-chain fatty acid (6:0 + 8:0 + 10:0 + 11:0 + 12:0); LCFA, long-chain fatty acid (14:0 + 14:1 + 15:0 + 15:1 + 16:0 + 16:1 + 17:0 + 17:1 + 18:0 + 18:1nt9 + 18:1nc9 + 18:2nt6 + 18:2nc6 + 20:0 + 18:3n6 + 20:1 + 18:3n3 + 21:0 + 20:2n6 + 22:0 + 20:3n6 + 22:1 + 20:3n3 + 23:0 + 20:4n6 + 22:2n6 + 24:0 + 20:5n3 + 24:1 + 22:6n3). SFA, saturated fatty acids (6:0 + 8:0 + 10:0 + 11:0 + 12:0 + 13:0 + 14:0 + 15:0 + 16:0 + 17:0 + 18:0 + 20:0 + 21:0 + 22:0 + 23:0 + 24:0 UFA, unsaturated fatty acids including MUFA: monounsaturated fatty acids and PUFA: polyunsaturated fatty acids; MUFA: (14:1 + 15:1 + 16:1 + 17:1 + 18:1n9t + 18:1n9c + 20:1 + 22:1 + 24:1); PUFA: (n-6 PUFA, n-6 polyunsaturated fatty acids + n-3 PUFA, n-3 polyunsaturated fatty acids); n-6PUFA: (18:2nt6 + 18:2nc6 + 18:3n6 + 20:2n6 + 20:3n6 + 20:4n6 + 22:2n6); n-3PUFA: (18:3n3 + 20:3n3 + 20:5n3 + 22:6n3); n-6 LCPUFA, n-6 long chain polyunsaturated fatty acids (20:2n6 + 20:3n6 + 20:4n6 + 22:2n6); n-3 LCPUFA, n-3 long chain polyunsaturated fatty acids (20:3n3 + 20:5n3 + 22:6n3); n-6/n-3, n-6 PUFA/n-3 PUFA; U/S; UFA/SFA; P/S, PUFA/SFA; IA, index of atherogenicity; IT, index of thrombogenicity.

**Table 5 animals-15-00423-t005:** Biochemical parameters of the serum of lactating donkeys fed alfalfa hay with a partial replacement of low-quality roughage.

	Treatment			
Items	LG	HG	SEM	*p*-Value
CHO (mmol/L)	2.02	1.66	0.101	0.031
D_3_HB (mmol/L)	0.26	0.23	0.012	0.078
TG (mmol/L)	0.62	0.58	0.033	0.461
HDL-C (mmol/L)	0.64	0.62	0.058	0.812
LDL-C (mmoL/L)	0.14	0.07	0.009	<0.01
NEFA (umol/L)	126.42	128.31	3.782	0.753

Abbreviations: LR, low alfalfa hay group, roughage ratio of 40 concentrate/60 roughage, alfalfa hay: 44.85 g/kg dry matter; HR, high alfalfa hay group, roughage ratio of 30 concentrate/70 roughage alfalfa hay: 179.48 g/kg dry matter; CHO, cholesterol; D_3_HB, β-hydroxybutyric acid; TG, triglycerides; HDL-C, high-density lipoprotein cholesterol; LDL-C, low-density lipoprotein cholesterol; NEFA, non-esterified fatty acids.

## Data Availability

The data presented in this study are available upon request from the corresponding author.

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
