# Peer review of "The Effect of Increasing the Proportion of Dietary Roughage Based on the Partial Replacement of Low-Quality Roughage with Alfalfa Hay on the Fatty Acid Profile of Donkey Milk"

_animals, 2025, doi:10.3390/ani15030423_

Round 1
Reviewer 1 Report
Comments and Suggestions for Authors
The manuscript ID animals-3344855 entitled “Effect of increasing the proportion of dietary roughage based on the replacement of some low-quality roughage by alfalfa on fatty acid profile of donkey milk” submitted to Animals as a research article, deals with the effects of two dairy donkey diets (isoenergetic and isonitrogenous) on the fatty acid profile studied in milk and blood.
According to the Journal aims, the paper deserves attention because of its potential relevancy in animal sciences as regard to the effects of forage quality (the dietary basis in hindgut fermenters) on milk yield and characteristics and plasma nutritional profile of lactating jennies.
In my opinion, the topic is very interesting, and this study is original, but the paper fails to properly describe the study aim and methods, including feeding strategy and husbandry, so that discussion . Moreover, the scientific terminology used throughout the text is too often omitted.
In more detail, my major concerns:
INTRODUCTION: A greater focus should be devoted to the main topic of the paper, which is the role of forages in the dairy donkey diet, as related to animal welfare, milk production, and milk (and plasma) fatty acids profile.
MATERIAL AND METHODS: in my opinion this section is very confusing.
- Were jennies clinically healthy?
- Did you weigh jennies at the end of the experiment?
- Did you score the body condition at d0 and at the end of the experiment? How about the fecal consistency?
- l. 105: Do you mean “parity” instead of “foetal numbers”?
- l. 106-108: Details of the experimental groups are omitted
- l. 108-110: the study compared two isoenergetic and isonitrogenous diets, where concentrate to forage ratio was 40:60 in LG and straw accounted for 73.6% of DM from forages while in HG, having a concentrate to forage ratio 30:70, straw accounted for 25.83 %DM from forages. This is not a simple replacement of “some low quality roughage” in the diet. Therefore, the straw content in LG vs. HG diet should be properly presented and, later, discussed. Was alfalfa administered as fresh forage or hay? Please specify. Was the lignin content measured?
- Table 1:
- erase “as fed” after DM.
- l. 125: Vitamin A, Vitamin D, Vitamin E instead of VA, VD, VE.
- l. 126 -141: to be moved to M&M section (not as footnote)
- l. 126-128: DE is digestible (not digestive) energy. Please note that UFC is the value of NET ENERGY (for HORSES) for feedstuffs (and requirements) as related to the net energy value of one kg (as fed) of a reference barley, used for maintenance in the horse. Therefore UFC unit is n/kg.
- l. 128-129 this sentence is not clear.
- l. 144: 3.5 m x 2 m cubicle for each jenny (and foal, I suppose) but (from l 114-115) one paddock (3.5m x 2 m) for all jennies separated from that for foals (1.6m x 2 m)? Ground inside and outside the paddock? That is confusing: were jennies individually or group fed? Please specify. If group fed, they cannot be considered independent observations.
- l. 150: please give the essential components of the milking plant. Were jennies adapted to the machine milking?
- l. 156: did you measure the DM content? Methods?
- l. 177-192: should precede text in l.168-176. How was the freeze drying carried out? Please provide details on FA standard used.
- l. 194: do you mean “from fasting animals?
- l. 196: how a tube can puncture?
- l. 207-217: “Diet samples and analyses” should be moved before “Milk sampling and analyses”
- l. 210-215: please specify in brackets AOAC, before the quoted Methods
- l. 215-217: please give reference for the FA intake
- l. 219: I suggest “performed by” instead of “conducted in”.
Table 2 – Dry matter and FA intake should be separated by milk yield and fat content. Saturated FA, mg/(kg?)W0.75 x d, please check.
DISCUSSION: the interesting results on DMI and milk yield (which is P<0.05) are not sufficiently discussed with literature.
l. 321: wheat straw?
l. 333-336: because of the well-known difference between ruminants and monogastric herbivores, discussion on cow milk should be limited.
I suggest reorganising text on 1. DMI, 2. milk production and investigated characteristics, 3. jennies’ nutritional profile, considering the straw content (as replaced by alfalfa) in the diet for hindgut fermenters.
CONCLUSION: Should be revised accordingly.
TITLE, ABSTRACT: Should be revised accordingly.
Comments on the Quality of English LanguageSufficient. Expect for typing mistakes and omitted scientific terminology.
Author Response
Reviewer #1:
1) INTRODUCTION: A greater focus should be devoted to the main topic of the paper, which is the role of forages in the dairy donkey diet, as related to animal welfare, milk production, and milk (and plasma) fatty acids profile.
Response: Thank you for your suggestions, we have revised the entire text to keep the content on topic.
MATERIAL AND METHODS:
2) Were jennies clinically healthy?
Response: We used clinically healthy jennies, and all experimental procedures performed on the jennies were conducted according to the national standard "Guidelines for Ethical Approval Document for Biomedical Research at Inner Mongolia Agricultural University. The experimental animals involved in this experimental project were examined by the Laboratory Animal Welfare and Ethics Committee of Inner Mongolia Agricultural University, which conformed to ethical principles and agreed to the program. Please see line 95-100.
3) Did you weigh jennies at the end of the experiment?
Response: At the end of the experiment, we weighed the jennies. We weighed the jennies on a regular weekly fasting basis during the data collection period, but according to the statistical results, the weight loss in lactation of the females did not differ between the two groups and as our main focus was on the effect of roughage on milk fatty acids, we did not present this part of the data, which will be published in a subsequent study.
4) Did you score the body condition at d0 and at the end of the experiment? How about the fecal consistency?
Response: We did not score the jennies for body condition throughout the trial, and we did not identify the consistency of the jennies' faeces at the end of the trial, but it was sensually evident that the faecal profile presented a relatively smooth appearance, brown or dark brown oval shape.
5) l. 105: Do you mean “parity” instead of “foetal numbers”?
Response: We have expressed here the number of pregnancies, which has been changed to “parity”, please see line 102.
6) l. 106-108: Details of the experimental groups are omitted
Response: Corrections have been made, please see lines 103-105 for details.
7) l. 108-110: the study compared two isoenergetic and isonitrogenous diets, where concentrate to forage ratio was 40:60 in LG and straw accounted for 73.6% of DM from forages while in HG, having a concentrate to forage ratio 30:70, straw accounted for 25.83 %DM from forages. This is not a simple replacement of “some low quality roughage” in the diet. Therefore, the straw content in LG vs. HG diet should be properly presented and, later, discussed. Was alfalfa administered as fresh forage or hay? Please specify. Was the lignin content measured?
Response: Thank you for the suggestion that alfalfa was fed as hay without measuring lignin content, see line 365-368 for a follow-up discussion on dietary roughage.
Table 1:
8) erase “as fed” after DM.
Response: Thanks for the suggestion, it has been removed, see the Table 1.
9) l. 125: Vitamin A, Vitamin D, Vitamin E instead of VA, VD, VE.
Response: Thank you for the correction, these errors have been corrected as detailed in line 126.
10) l. 126 -141: to be moved to M&M section (not as footnote)
Response: Thank you for your suggestion, this is detailed in M&M (line 107-110 171-183).
11) l. 126-128: DE is digestible (not digestive) energy. Please note that UFC is the value of NET ENERGY (for HORSES) for feedstuffs (and requirements) as related to the net energy value of one kg (as fed) of a reference barley, used for maintenance in the horse. Therefore UFC unit is n/kg.
Response: Because the nutritional requirements and feeding standards of donkeys were very limited, we chose to use the UFC only as a reference, and there was little data on metabolisable energy and heat increment in donkeys in the previous studies, we chose to use digestible energy as the energy value of the ration.
12) l. 128-129 this sentence is not clear.
Response: Thank you for the reminder. Changes have been made, as detailed in lines 127-128.
13) l. 144: 3.5 m x 2 m cubicle for each jenny (and foal, I suppose) but (from l 114-14) one paddock (3.5m x 2 m) for all jennies separated from that for foals (1.6m x 2 m)? Ground inside and outside the paddock? That is confusing: were jennies individually or group fed? Please specify. If group fed, they cannot be considered independent observations.
Response: Each donkey is fed separately and each donkey's paddock is separate from the foal's paddock.
14) l. 150: please give the essential components of the milking plant. Were jennies adapted to the machine milking?
Response: The milking machine is made up of two parts: the milking apparatus and the vacuum device. The selection of a suitable milking apparatus for the size and shape of the donkey's teats, and the use of the suction effect produced by the vacuum device to simulate the sucking action of the donkey foal, sucking out the milk. The first week of the trial was a transition period just to allow the donkeys to adapt to the milking equipment, and the donkeys adapted well to the milking equipment.
15) l. 156: did you measure the DM content? Methods?
Response: Yse, we did. Please see line 156-158.
16) l. 177-192: should precede text in l.168-176. How was the freeze drying carried out? Please provide details on FA standard used.
Response: milk samples were collected from each donkey and a portion of the milk samples collected twice that day were mixed and poured evenly into a stainles steel tray, frozen at -80°C and transferred to a freeze dryer overnight. The next day, the samples were freezedrying in a Sublimator 3 × 4 × 5 freeze dryer (Zirbus Technology, Bad Grunt, Germany). The samples were transferred to the freeze dryer operating at a maximum temperature of 35 â—¦C in a condensation chamber under vacuum at a maximum pressure of 0.3 mbar. Freeze-drying was completed in 72 h. The powders obtained were stored under vacuum in sealed polyethylene bags for further analysis.
Information on the 37 mixed fatty adipic acid standards used and absorption peak profiles are detailed in the Supplementary Material.
17) l. 194: do you mean “from fasting animals?
Response: On the morning of the sampling day, we do not feed the donkeys any rations before collecting the blood, and the donkeys are fasted, because the collection of blood after feeding the donkeys rations may affect the fluctuation of some of the biochemical indices of the blood.
18) l. 196: how a tube can puncture?
Response: Blood is collected by puncture using a disposable sterile blood collection needle. See line 223-225 for details.
19) l. 207-217: “Diet samples and analyses” should be moved before “Milk sampling and analyses”.
Response: Thank you for your suggestion, which has been amended accordingly, as detailed in line 163-184.
20) l. 210-215: please specify in brackets AOAC, before the quoted Methods
Response: Changes have been made, as detailed in line 156 and 166-171.
21) l. 215-217: please give reference for the FA intake
Response: Due to the limited number of studies reported on the nutritional needs and feeding standards of jennies, there are no sound and exhaustive data on this area of research for the time being, which will be followed up further.
22) l. 219: I suggest “performed by” instead of “conducted in”.
Response: Thank you for your suggestion, changes have been made as detailed in line 243.
23) Table 2 – Dry matter and FA intake should be separated by milk yield and fat content. Saturated FA, mg/(kg?)W0.75 x d, please check.
Response: Thanks for the correction, which has been made, as detailed in Table 2.
24) DISCUSSION: the interesting results on DMI and milk yield (which is P<0.05) are not sufficiently discussed with literature.
Response: A detailed discussion of the results on feed intake and milk production in the two treatment groups has been published for review in Anim. Feed Sci. Technol. (2022, 292, 115444. https://doi.org/10.1016/j.anifeedsci.2022.11544).
25) l. 321: wheat straw?
Response: Thank you for the correction, which has been made, as detailed in line 353.
26) l. 333-336: because of the well-known difference between ruminants and monogastric herbivores, discussion on cow milk should be limited.
I suggest reorganising text on 1. DMI, 2. milk production and investigated characteristics, 3. jennies’ nutritional profile, considering the straw content (as replaced by alfalfa) in the diet for hindgut fermenters.
Response: Thank you for your suggestion, it has been rewritten, please see line 369-373, and add references [28] and [29].
27) CONCLUSION: Should be revised accordingly.
Response: Thanks for the suggestion, it has been rewritten, see conclusion for details (line 437-440).
28) TITLE, ABSTRACT: Should be revised accordingly.
Response: Thanks for the suggestion, it has been rewritten, see TITLE, ABSTRACT for details(Changes in this section have been highlighted in red).
Reviewer 2 Report
Comments and Suggestions for Authors
Your paper is interesting and relevant. Please review the English, the acronyms, and the organization of the tables.

English needs revision.
Author Response
Reviewer #2:
1) Line 22. Regarding the acronym N-3, please correct all instances of the acronym and standardize it in the text as n-3.
Response: Thanks for the correction, it has been changed, please see line 22, 44 and 50.
2) Standardize the acronyms FA and PUFA throughout the text after lines 42 and 44
Response: Thanks for the correction., it has been changed, please see line 66, 80, 84, 85, 86, 183-184, 211, 250, 257, 303, 346, 348, 364, 392, 393, 395, 421, 422.
3) Line 24, 32, 88.“ The phrase "... ratio of dietary concentrate to roughage" is unclear.
Please revise the English for clarity. I suggest the term "Concentrate:roughage" as it may provide more clarity.
Response: Thank you for your suggestion, it has been changed, please see line 24, 32, 33, 83.
4) Line 32: The sentence “… (LG, concentrate to forage ratio of 40:60) and a high alfalfa feed group (HG, concentrate to forage ratio of 30:70)” is unclear. Please rewrite it for clarity. For example: (LG, roughage ratio of 40 concentrate:60 alfalfa hay or fresh) and a high alfalfa feed group (HG, roughage ratio of 30 concentrate:70 alfalfa hay).Also, clarify whether it was alfalfa hay, haylage, or fresh alfafa. Please review the description in the methodology (lines 107 to 110) and in the abstract.
Response: Thank you for your suggestion, it has been changed, please see line 32-33 and 103-105.
5) Line 37. Keywords: donkey; fatty acid; milk; plasma; roughage. Do not repeat words from the title.
Response: Thank you for your suggestion, it has been changed, please see line 38.
6) Line 58-59: “…when n-6 / n-3 is less than 5, it can effectively prevent the occurrence of cancer [7].” Please verify the citation for reference [7]. The information does not appear to be found in that article
Response: Thanks for the correction, it has been changed, please see line 52.
7) Lines 100–103: Please rewrite for clarity. Where was the study conducted? Was it at the Experimental Farm of Inner Mongolia Agricultural University or at the Yulv Donkey Grassland Farm (Hohhot, China)?
Response: Thanks for the correction, it has been changed, please see line 95.
8) Line 105. “…foetal number (3.25 ± 1.12 foetuses)...” What does this mean?
Response: We have expressed here the number of pregnancies, which has been changed to “parity”, please see line 102.
9) Line 111-112: Please replace "pre-feeding period" with "adaptation period to the diet" and "experimental period" or "data collection period".
Response: Thanks for the correction, it has been changed, please see line 111.
10) Table 1:
Please describe the acronyms LG and HG: Low Alfalfa Group (LAG) and High Alfalfa Group (HAG).
- Alfalfa hay: Clarify the type of alfafa used (e.g., hay, haylage, or green alfalfa).
- The relationship 40:60 and 30:70 is unclear. Does it refer to 60% and 70% alfalfa hay?
Please clarify .
- Highlight the types of foods: concentrate and roughage.
- Provide descriptions for abbreviations: U/S, P/S, LCPUFA, etc.
Response: Thank you for your suggestion, Table 1 has been revised as detailed in the labeled section.
11) 2.3. Milk Sampling and Analysis
Include the descriptive paragraph on milking as mentioned in item 2.2 Feeding Management:
Line 147 – 151. “Donkeys were separated from their foals for 3 hours each time from 7:00 to 10:00 a.m. and 2:00 to 5:00 p.m. for a total of 6 hours per day for milk collection. Milking was performed daily at 10:00 a.m. and 5:00 p.m. using a 149 separate vacuum milk extractor (Judu-H5402, Judu Technology Co. Ltd., Hebei, China), and milking was measured and recorded using a lactation meter connected to the milking machine.
Were the samples collected by milking? Were 2 samples per day? Was the fatty acid analysis conducted on a composite sample? How was the sample composed for fatty acid analysis? For the composite sample, was it collected per animal, 2 samples per day, and 16 samples per week?
What is the average percentage of fatty acids in the composite sample (weekly/animal), considering a mean of 7 samples over 7 weeks of milk collection for sampling? Were the samples from the first week of collection used, or was the first week considered as the diet adaptation period? Please explain in detail.
Response: The samples were collected throughout the trial period. Milk samples from each donkey were collected on the last day of the eighth week respectively, and the morning and afternoon samples were mixed in a 1:1 ratio. When collecting samples, each donkey was taken as a replicate, and each replicate was collected once. A total of 16 samples were collected.
12) Line 193 – 195. 2.4. Blood Sampling and Analysis
At the end of the experiment, blood was collected on an empty stomach in the morning before feeding the donkeys.
Clarify: Only one sample collection was performed on day X of the experimental period. This involved 1 collection, n animals, and 2 treatments.
Response: On the last day of the trial, blood samples from each donkey was collected respectively on an empty stomach in the morning before feeding the donkeys,and a total of 16 blood samples from 2 treatment groups were collected. Considering that blood collection may stress donkeys, which may affect milk composition or milk production, the collection of blood samples was collected on the last day of the data collection period (milk samples were collected the day before) to reduce the impact of blood collection stress on the test.
13) Statistical Analysis – Proofread the text
Line 221-223> “... differences were considered significant at P ≤ 0.05 and differences were considered statistically significant at 0.05 < P ≤ 0.10.”
Response: Thank you for your suggestion, it has been corrected, see line 246 for details.
14) Table 5. Please adjust the values of SEM and P-value to 3 decimal digits.
Response: Thank you for your suggestion, it has been corrected, see Table 5.
15) For all tables:
-Revise the titles: Correct any errors in the titles. Write the titles in full and ensure they clearly describe the content of the table.
-Include the top row: Make sure the top row specifies the treatment (e.g., Table 1).
-Include units: Ensure that units are included for all relevant variables in the tables.
-Standardize the acronyms for treatments: Review and standardize the acronyms for the treatments both in the text and in the tables. There seems to be confusion regarding the use of acronyms, so it is important to be consistent across the document.
Response: Thanks for the correction. Changes have been made to all tables throughout the text. See Tables 1-5 for details.
16) Line 318: Please change “mares” (equine females) to “jennies” (female donkeys).
Response: Thanks for the correction, it has been changed, please see line 349.
17) Line 333 – 337: Is this possibility of pre-cecal absorption of fatty acids from roughage in nonruminants described in any article?
Response: Unfortunately, due to the limited progress of current research, no relevant articles could be found, so the formulation of this section has been modified, as detailed in lines 373-379, and add reference [29], [30], [31].
18) Line 338-340: Please rewrite the paragraph, as is unclear.
Response: Thank you for your suggestion, it has been corrected, see line 380-381.
19) Line 351: Ration or diet?
Response:Diet, corrected, please see line 393.
20) Line 366: Please rewrite the following paragraph, as is unclear.
The n-3 PUFA content of donkey milk was 7.97 g and ALA content was 7.25 g per 100 g of milk fat; the n-3 PUFA content of cow milk was 0.78 g and ALA content was 0.48 g [10], and the ALA content of donkey milk fat was 15 times higher than that of cow milk. Donkey milk has a higher content of unsaturated fatty acids, especially ALA, than cow's milk, and therefore remains the main dairy source of n-3 PUFA for the human body. According to previous reports, the content of ALA in cow's milk fat was found to be about 0.29 – 0.79g [35, 36, 37], and in combination with the present experimental study, the content of ALA in donkey's milk fat was found to be about 3.13 – 8.76g [10, 38, 39]. In this study, we found that alfalfa substitution of some low-quality roughage and increasing the percentage of roughage in the diet from 60% to 70% significantly reduced the concentration of CHO and LDL-C in donkeys’ serum and tended to reduce the content of D3HB. Higher proportion of n-6 / n-3 in the diet also acts as a potential risk factor for cardiovascular disease [40].
Did you calculate the Atherogenicity and Thrombogenicity indices for each treatment? I suggest including this information.
Response: Thank you for your suggestion, we have rewritten this section and added these two indicators, statistics, discussion of results and analyses as detailed in lines 428-435.
21) In this study, it was concluded that the increasing the proportion of alfalfa fed in the roughage of lactating donkeys’ diet, increased the percentage of all the above functional FAs in donkey milk and decreased the ratio of n-6 / n-3; the content of C14:0 and C16:0 in the LG was not significantly different from that of the HG. This indicates that the quality and proportion of roughage has an important effect on the fatty acid composition of donkey milk.
Response: Thank you for the correction, it has been revised as requested, see 413-423 for details.
22) Please rewrite: “…, and that the FAs composition and value (???) of donkey milk will be optimized after reducing the level of concentrate and replacing low quality roughage with alfalfa.
Response: Thanks for the correction, it has been deleted “value”. Please see line 422.
23) Lines 391-395. Conclusions
The alfalfa (hay or fresh??) substitution of low-quality roughage to increase the percentage of dietary roughage in10%, could increase the ALA fatty acid content, tending to increase the n-3 PUFA and LCFA content, and decrease the n-6/n-3 of donkey milk. Those optimized the milk fatty acid quality.
Response: Thanks for the correction, it has been made. Please see line 437.
24) Revise the English: Please review and revise the English throughout the document for clarity.
Response: Thank you for your suggestions, we have made changes and corrections throughout the text.
25) Standardize the acronyms for treatments. Ensure that the acronyms for the treatments are consistent throughout both the text and tables.
Response: Thank you for your suggestions, we have made changes and corrections throughout the text.
26) Clarify the composition of the diets: Make the composition of the diets clearer, especially the concentrate:roughage ratio, and include relevant details.
Response: Thank you for your suggestions, we have made changes and corrections throughout the text.
27) Include the Atherogenicity (AI) and Thrombogenicity (TI) indices: Calculation of the atherogenicity index (AI) and thrombogenicity index (TI) in the tables and discussion. These can be calculated using the equation described by Ulbricht and Southgate (1991):
AI = (12:0+(4×14:0)+16:0)(12:0 + (4 × 14:0) + 16:0)(12:0+(4×14:0)+16:0) / ΣMUFA+Σ(n−6)+Σ(n−3)ΣMUFA + Σ(n−6) + Σ(n−3)ΣMUFA+Σ(n−6)+Σ(n−3)
TI = (14:0+16:0+18:0)(14:0 + 16:0 + 18:0)(14:0+16:0+18:0) / 0.5×ΣMUFA+0.5×Σ(n−6)+(3×Σ(n−3))+(Σ(n−3)/Σ(n−6))0.5 × ΣMUFA + 0.5 × Σ(n−6)
+ (3 × Σ(n−3)) + (Σ(n−3) /Σ(n−6))0.5×ΣMUFA+0.5×Σ(n−6)+(3×Σ(n−3))+(Σ(n−3)/Σ(n−6))
where ΣMUFA = sum of monounsaturated fatty acids.
Response: Thank you for your suggestion, we have rewritten this section and added these two indicators, statistics, discussion of results and analyses as detailed in lines229-235; 409-413, and add reference [22] and [31].
28) Review the titles, footnotes, and tables structure:
Please review and ensure consistency in the titles, footnotes, and overall structure of the tables.
Response: Thank you for your suggestions, we have made changes and corrections throughout the text.
Round 2
Reviewer 1 Report
Comments and Suggestions for Authors
The Authors have made some modifications on the manuscript ID animals-3344855 entitled “Effect of increasing the proportion of dietary roughage based on the replacement of some low-quality roughage by alfalfa on fatty acid profile of donkey milk” and they sent responses to some doubts and concerns.
Unfortunately, the Authors did not modify the Introduction by highlighting that this paper is a further analysis of Liang et al [16]. In my opinion, this is essential for the reader to understand the paper value. However, it should be noted that milking yield differs in the two papers. The Authors also state (response #3 and 4) the unavailability for this paper of the results on body weight (and condition) variations, notwithstanding their possible effects on plasma and milk FA reported for equines in literature.
Besides these major concerns, please note that throughout the text, the adjective “partial” referred to "replacement" sounds more scientific than “some”. Moreover,
L. 1-4. Suggested title: “Effect of increasing the proportion of dietary roughage based on the partial replacement of low-quality roughage by alfalfa hay on fatty acid profile of donkey milk”.
L. 107-109. As I wrote in my previous review of the paper, I invite the Authors to modify the text as DE means digestible energy not digestive. I would suggest modifying the sentence as follows: “The energy value, expressed as digestible energy (DE), was calculated using the equations reported by INRA (2015).”
L. 127. again, please use "digestible" instead of "digestive"
L. 128-129. to be moved in M&M
L. 149. please give details on machine-milking vacuum pump (flow rate) and parameters, like vacuum level, pulsation ratio, and pulsation rate.
L. 150. Please confirm the premilking use of ethanol 70%
L. 184 "weight" instead of "weighe"
L. 189-190. please modify “Milk samples from each donkey were also collected…”
L. 197. please add “for FA analysis”
L. 221. I still suggest “from fasting animals” instead of “on an empty stomach”
L. 229 and 299. Please modify “atherogenic” instead of “atherosclerosis” and “thrombogenic” instead of “thrombosis”
TABLE 2.
- I still remark that dry matter intake cannot be considered lactation performance.
- please correct “mg/kg W0.75 × day”
L. 374-376. redundant. Please check the whole sentence for consistency with L. 357-359.
I confirm the interest of the proposed manuscript as further in-depth analysis on donkey milk but in my opinion significant improvements are still needed to make the paper worth to be published in Animals.
Author Response
Dear Reviewer
Thank you for your letter and constructive comments concerning our manuscript entitled “Effect of increasing the proportion of dietary roughage based on the replacement of some low-quality roughage by alfalfa on fatty acid profile of donkey milk”. We have studied your comments carefully and made moderate correction which we hope meet with your approval. We answer your questions or comments in details in the following texts. Using red to highlight revise made by the reviewer, while blue marks indicate rephrasing to reduce repetition.
Reviewer #1
1) Unfortunately, the Authors did not modify the Introduction by highlighting that this paper is a further analysis of Liang et al [16]. In my opinion, this is essential for the reader to understand the paper value. However, it should be noted that milking yield differs in the two papers. The Authors also state (response #3 and 4) the unavailability for this paper of the results on body weight (and condition) variations, notwithstanding their possible effects on plasma and milk FA reported for equines in literature.
Response: Thank you for your comments. In the introductory section, we have modified “The results of our previous studies found...” and added “Therefore, based on our previous studies” to emphasize that this article is a further analysis of our previous studies. Please see line 77-83 and 85.
Due to an oversight on my part, I wrote the wrong values related to the milking yield indicator in Table 2, which has now been corrected and the source of the data has been noted in a footnote to Table 2. In addition, we have added the body weight loss indicator to Table 2 and updated the description accordingly in the Materials and Methods (line 156-160 and 246), Results (line 254), and Discussion (line 389-395) sections.
2) Besides these major concerns, please note that throughout the text, the adjective “partial” referred to "replacement" sounds more scientific than “some”. Moreover, L. 1-4. Suggested title: “Effect of increasing the proportion of dietary roughage based on the partial replacement of low-quality roughage by alfalfa hay on fatty acid profile of donkey milk”.
Response: Thank you for your suggestion. Following your suggestion, we revised the title to “Effect of increasing the proportion of dietary roughage based on partial low-quality roughage by alfalfa hay on fatty acid profile of donkey milk”.
3) L. 107-109. As I wrote in my previous review of the paper, I invite the Authors to modify the text as DE means digestible energy not digestive. I would suggest modifying the sentence as follows: “The energy value, expressed as digestible energy (DE), was calculated using the equations reported by INRA (2015).”
Response: Thank you for your suggestion, we have made a replacement here, please see line107-108.
4) L. 127. again, please use "digestible" instead of "digestive"
Response: Thank you for your suggestion, we have made a replacement here, please see line126.
5) L. 128-129. to be moved in M&M
Response: Thank you for the correction, prior L. 128-129 has been removed, the expression Neutral detergent fiber (NDF) was tested with heat-resistant amylase and expressed exclusive of residual ash. Acid detergent fiber (ADF) expressed ‘’ has been updated in Materials and methods, please see line 108-110, and the serial numbers of the footnotes to Table 2 have been updated.
6) L. 149. please give details on machine-milking vacuum pump (flow rate) and parameters, like vacuum level, pulsation ratio, and pulsation rate.
Response: The vacuum pump has a vacuum level of around 50 KPa, a pulsation frequency of 60 ± 3 times per minute and a pulsation ratio of 60:40. Please see line 146-148.
7) L. 150. Please confirm the premilking use of ethanol 70%.
Response: Thanks for the correction, I wrote the wrong alcohol concentration due to an oversight on my part. It should be 75% alcohol here, see line 149 for details.
8) L. 184 "weight" instead of "weighe".
Response: Thank you for the correction, which has been made, as detailed in line 186.
9) L. 189-190. please modify “Milk samples from each donkey were also collected…”
Response: Thank you for the correction, “also” has been added here, please see line 191.
10) L. 197. please add “for FA analysis”.
Response: Thank you for the correction, which has been made, as detailed in line 199.
11) L. 221. I still suggest “from fasting animals” instead of “on an empty stomach”.
Response: Thank you for the correction, which has been made, as detailed in line 223.
12) L. 229 and 299. Please modify “atherogenic” instead of “atherosclerosis” and “thrombogenic” instead of “thrombosis”
Response: Thank you for the correction, We decided to write the indicators in a format consistent with that of the cited references based on the references given by reviewer 1. We referred to the references for the final revision to revise the expressions IA and IT to “ index of atherogenicity and index of thrombogenicity”. And revised the relevant descriptions in Materials and Methods. Please see line 230-231.
13) L. TABLE 2.
- I still remark that dry matter intake cannot be considered lactation performance.
- please correct “mg/kg W0.75 × day”
Response: Thanks to your correction, we have modified the title of Table 2 into production performance and fatty acid intake and modified the units of fatty acid intake as detailed in Table 2. Please see the Table 2.
14) L. 374-376. redundant. Please check the whole sentence for consistency with L. 357-359.
Response: Thank you for the suggestion, previous lines 374-376 were deleted. Please see line 380.
Reviewer 2 Report
Comments and Suggestions for Authors
Dear Authors,
The article has improved after the first revision, but I still have some suggestions and questions.
1. English needs to be reviewed.
2. I suggest rewriting the title for better clarity.
3. Standardize table titles and table structure. Tables must be individually accessible, so their titles and footnotes must be complete and informative.
The structure of the tables has not been revised. There is a suggestion in the attached file.
4. The article objective: Please, keep it short and clear.
“This study mainly investigated the effects of substituting partial millet straw with alfalfa to increase the proportion of roughage on the changes of plasma and milk FA profiles and blood biochemical parameters in lactating donkeys, aiming to provide a theoretical basis for rational allocation of roughage and optimization of FAs profile of lactating donkey milk.”
The last sentence doesn't make sense and isn't part of your goal.
Suggestion:
“This study mainly investigated the effects of substituting partial low-quality roughage with high-quality hay and increasing the roughage proportion on the plasma and milk FA profile changes in lactating donkeys.
5. Review the hypothesis, objective, and conclusion.
The effect of the replacement of straw with alfalfa and the effect of increased roughage proportion are confounded, therefore, the results are a consequence of both effects. That must be clear in the text. For instance, did the n3 content increase because alfalfa was included? Or did the n3 content increase because the total amount of roughage increased by 10%?
6. How much did n3 increase in milk with the 10% higher-quality roughage in the total diet?
7. Please, include the AI and TI indices in tables 2 and 4, as they are related to the food quality (milk for humans). In my opinion, these indexes in plasma do not make sense (Table 3). Did you read it in some articles?
According to Simopoulos (2002) a n−6/ n−3 ratio below 2.0 is highly desirable, reducing the risk of many chronic diseases.
Simopoulos, A. P. (2002). The importance of the ratio of omega-6/omega-3 essential fatty acids. Biomedicine & Pharmacotherapy, 56, 365–379. https://doi.org/10.1016/S0753 -3322(02)00253 -6.
Lines 409 to 413 – I am not sure about this Reference [38]. The article does not discuss the Atherogenicity and thrombogenicity indexes. There are publications with donkey milk that describe them.
Line 299 – Atherogenicity and thrombogenicity indexes – Correct according to the citation below:
Ulbricht, T. L. V., & Southgate, D. A. T. (1991). Coronary heart disease: Seven dietary factors. The Lancet, 338, 985–992. https://doi.org/10.1016/0140-6736(91)91846
Linha 221. “…was collected preprandial (x hours fasting) in the morning before feeding the donkeys.”

1. English needs to be reviewed.
2. I suggest rewriting the title for better clarity.
Author Response
Dear Reviewer
Thank you for your letter and constructive comments concerning our manuscript entitled “Effect of increasing the proportion of dietary roughage based on the replacement of some low-quality roughage by alfalfa on fatty acid profile of donkey milk”. We have studied your comments carefully and made moderate correction which we hope meet with your approval. We answer your questions or comments in details in the following texts. Using red to highlight revise made by the reviewer, while blue marks indicate rephrasing to reduce repetition.
Reviewer #2:
1) English needs to be reviewed.
Response: Thank you for your valuable and thoughtful comments. We have carefully checked and improved the English writing in the revised manuscript.
2) I suggest rewriting the title for better clarity.
Response: Thank you for the suggestion. Since reviewer 1 also made a revision to the title, and taking into account reviewer 1's revision. In order to summarize reviewer 1's comments, we have revised the title to “Effect of increasing the proportion of dietary roughage based on partial low-quality roughage by alfalfa hay on fatty acid profile of donkey milk”.
3) Standardize table titles and table structure. Tables must be individually accessible, so their titles and footnotes must be complete and informative.
Response: Thank you for your valuable and thoughtful comments. We have standardised the table titles and table structure in line with your suggestions and have updated the content of the footnotes to the table titles as well as the serial numbers, please see Tables 1-5 in red words.
4) The article objective: Please, keep it short and clear.
“This study mainly investigated the effects of substituting partial millet straw with alfalfa to increase the proportion of roughage on the changes of plasma and milk FA profiles and blood biochemical parameters in lactating donkeys, aiming to provide a theoretical basis for rational allocation of roughage and optimization of FAs profile of lactating donkey milk.”
The last sentence doesn't make sense and isn't part of your goal.
Suggestion:
“This study mainly investigated the effects of substituting partial low-quality roughage with high-quality hay and increasing the roughage proportion on the plasma and milk FA profile changes in lactating donkeys.
Response: Thank you for the suggestion, for our previous findings we have rewritten the presentation and, at your suggestion, modified the hypothesis and purpose of the experiment for clarity and short, as detailed in line 86-91. ,
5) Review the hypothesis, objective, and conclusion.
The effect of the replacement of straw with alfalfa and the effect of increased roughage proportion are confounded, therefore, the results are a consequence of both effects. That must be clear in the text. For instance, did the n3 content increase because alfalfa was included? Or did the n3 content increase because the total amount of roughage increased by 10%?
Response: High-quality roughage is the main source of PUFA in diets compared to concentrated feeds. Studies have shown that alfalfa has higher PUFA content than millet straw, so increasing the proportion of alfalfa hay in roughage or increasing the proportion of total roughage can increase the total PUFA content in the diet. In this experiment, the HG group consumed more n-3 PUFA and the n-3 PUFA spectrum in blood was increased, which provided more material basis for the mammary epithelial cells to synthesize LCFA directly from blood uptake. However, the exact mechanism still needs to be further explored. We have made appropriate modifications to the hypotheses, objectives and conclusions of this paper, as detailed in line86-91 and 459-461.
6) How much did n3 increase in milk with the 10% higher-quality roughage in the total diet?
Response: In this study (Table 2), n-3 PUFA was 2.65% in total methyl ester in the milk of the LG group and 3.85% in total methyl ester in the milk of the HG group, with a 45.28% increased in n-3 PUFA. In lyophilized donkey's milk, the content of n-3 PUFA was 91.43 mg/kg in the milk of the LG group and 123.59 mg/kg in the milk of the HG group, with an increase in n-3 PUFA by 35.17% n-3PUFA increased by 35.17%.
7) Please, include the AI and TI indices in tables 2 and 4, as they are related to the food quality (milk for humans). In my opinion, these indexes in plasma do not make sense (Table 3). Did you read it in some articles?
According to Simopoulos (2002) a n−6/ n−3 ratio below 2.0 is highly desirable, reducing the risk of many chronic diseases.
Simopoulos, A. P. (2002). The importance of the ratio of omega-6/omega-3 essential fatty acids. Biomedicine & Pharmacotherapy, 56, 365–379. https://doi.org/10.1016/S0753 -3322(02)00253 -6.
Lines 409 to 413 – I am not sure about this Reference [38]. The article does not discuss the Atherogenicity and thrombogenicity indexes. There are publications with donkey milk that describe them.
Response: Thank you for pointing this out, I have put IA and IT into Tables 2 and 4, for IA and IT in Table 3. I chose to keep them and not delete them in Table 3, because these two metrics are also frequently found in human blood as a response to blood lipid metabolism. Thank you for giving a new literature on the discussion in breast we have made a new discussion (n-6/ n-3, IA and IT) on line 434-443.
The previous literature [38] on lines 409 - 413 has been verified, and due to an oversight on my part it was misplaced, and has been deleted, please see line 426, and as a result of updating the new literature, we have adjusted the new literature numbers throughout the text in a timely manner as well.
8) Line 299 – Atherogenicity and thrombogenicity indexes – Correct according to the citation below:
Ulbricht, T. L. V., & Southgate, D. A. T. (1991). Coronary heart disease: Seven dietary factors. The Lancet, 338, 985–992. https://doi.org/10.1016/0140-6736(91)91846
Response: Thank you for the correction, which has been made, we referred to the references for the final revision to revise the expressions IA and IT to “index of atherogenicity and index of thrombogenicity”. And revised the relevant descriptions in Materials and Methods. Please see line 230-231.
9) Linha 221. “…was collected preprandial (x hours fasting) in the morning before feeding the donkeys.”
Response: Thank you for the correction, reviewer 1 also made a revision here which has been made, as detailed in line 223.
Based on what you have revised in the Word version, we have deleted the parts that needed to be revised that were marked yellow in the old version, and the words that needed to be added are shown in red in the new manuscript.
Round 3
Reviewer 1 Report
Comments and Suggestions for Authors
The Authors modified the manuscript ID animals-3344855 and they sent responses to my concerns.
However, I am afraid our communication does not work well. Starting from the title and throughout the text, the adjective “partial” should be referred to “replacement” not to straw, whose quality is unquestionably low.
Therefore, I invite again the Authors to modify the manuscript as follows:
TITLE : “Effect of increasing the proportion of dietary roughage based on the partial replacement of low-quality roughage with alfalfa hay on fatty acid profile of donkey milk”
L 25 “substitution of low-quality roughage with high-quality roughage….”
L 27-28 ”…..the effects of partial dietary replacement of low-quality roughage with alfalfa hay to increase…..
L 36 ”Partial replacement of low quality….”
L 89-90 “…the effects of partial substitution of low-quality roughage with….”
L 382-383 “alfalfa partially replacing low-quality roughage…”
L 437-438 “ ….the partial replacement of low-quality…”
L 459 “The partial replacement of low-quality…”
Besides these remarks, please note
L 67 from scratch???
L 149 did you really use 75% ethanol, during milking??? For animal welfare reasons, I would never use it in lactating animals, neither before nor after machine milking
L 149 “Milk yield was……” (instead of milking)
L 152 please use left-overs (or refusals) instead of litter
L 246 Capital letter is missing at the beginning of the sentence
L. 258. n6/n3 ratio and IA were....
Table 2. "loss" instead of "lose"
L, 371-372. I totally disagree: it depends on the concentrate. This sentence should be modified or erased.
In my opinion few but relevant improvements are still needed to make the paper worth to be published by Animals.
Author Response
Dear Reviewer
Thank you for your letter and constructive comments concerning our manuscript entitled “Effect of increasing the proportion of dietary roughage based on the partial replacement of low-quality roughage with alfalfa hay on fatty acid profile of donkey milk”. We have studied your comments carefully and made moderate correction which we hope meet with your approval. We answer your questions or comments in details in the following texts. Using red to highlight revise made by the reviewer, while blue marks indicate rephrasing to reduce repetition.
Reviewer #1
1) Starting from the title and throughout the text, the adjective “partial” should be referred to “replacement” not to straw, whose quality is unquestionably low. TITLE : “Effect of increasing the proportion of dietary roughage based on the partial replacement of low-quality roughage with alfalfa hay on fatty acid profile of donkey milk”
Response: Thank you for the correction, we have modified the title, please see line 2-4.
2) L 25 “substitution of low-quality roughage with high-quality roughage….”
Response: Thank you for the correction, this has been revised, please see line 25-26.
3) L 27-28 “….the effects of partial dietary replacement of low-quality roughage with alfalfa hay to increase…”.
Response: Thank you for the correction, this has been revised, please see line 27-28.
4) L 36 “Partial replacement of low quality….”
Response: Thank you for the correction, this has been revised, please see line 36.
5) L 89-90 “…the effects of partial substitution of low-quality roughage with….”
Response: Thank you for the correction, this has been revised, please see line 89-90.
6) L 382-383 “alfalfa partially replacing low-quality roughage…”
Response: Thank you for the correction, this has been revised, please see line 381-382.
7) L 437-438 “ ….the partial replacement of low-quality…”
Response: Thank you for the correction, this has been revised, please see line 437-438.
8) L 459 “The partial replacement of low-quality…”
Response: Thank you for the correction, this has been revised, please see line 459.
9) L 67 from scratch???
Response: Thank you for the correction. This has been modified to mean “de novo”, please see line 67.
10) L 149 did you really use 75% ethanol, during milking??? For animal welfare reasons, I would never use it in lactating animals, neither before nor after machine milking.
Response: Terribly sorry, due to an oversight on my part I wrote here incorrectly that warm water was used for wiping the breasts, this has now been modified, please see line 149.
11) L 149 “Milk yield was……” (instead of milking)
Response: Thank you for the correction, this has been revised, please see line 149.
12) L 152 please use left-overs (or refusals) instead of litter
Response: Thank you for the correction, this has been revised, please see line 152-153.
13) L 246 Capital letter is missing at the beginning of the sentence
Response: Thank you for the correction, this has been revised, please see line 246.
14) L. 258. n6/n3 ratio and IA were....
Response: Thank you for the correction, this has been revised, please see line 258-259.
15) Table 2. "loss" instead of "lose"
Response: Thank you for the correction, this has been revised, please see Table 2.
16) L, 371-372. I totally disagree: it depends on the concentrate. This sentence should be modified or erased.
Response: Thank you for the suggestion, this has been deleted, please see line 371.
Reviewer 2 Report
Comments and Suggestions for Authors
Dear Authors,
The article has improved after the last revision, but I still have some suggestions.
1. Review the format of the in-text citations.
In general, the document is suitable for publication. Congratulations.
Author Response
Dear Reviewer
Thank you for your letter and constructive comments concerning our manuscript entitled “Effect of increasing the proportion of dietary roughage based on the partial replacement of low-quality roughage with alfalfa hay on fatty acid profile of donkey milk”. We have studied your comments carefully and made moderate correction which we hope meet with your approval. We answer your questions or comments in details in the following texts. Using red to highlight revise made by the reviewer, while blue marks indicate rephrasing to reduce repetition.
Reviewer #2
1) Review the format of the in-text citations.
Response: Thank you for your comments, we have checked and revised the in-text citation formatting, please see line 108, 171, 202, 232, 439, 442.